# Recombination and Mutation in a New HP-PRRSV Strain (SD2020) from China

**DOI:** 10.3390/v15010165

**Published:** 2023-01-05

**Authors:** Yang Xia, Tianying Zhang, Dengmei Gong, Juan Qi, Shenghai Jiang, Hao Yang, Xianchang Zhu, Yu Gan, Yi Zhang, Yanyan Han, Yan Li, Jida Li

**Affiliations:** 1Institute of Zoonosis, College of Public Health, Zunyi Medical University, Zunyi 563000, China; 2Southwest Guizhou Vocational & Technical College Nationalities, Southwest Guizhou Autonomous Prefecture, Xingyi 562400, China; 3Shijiazhuang Fuli Properties Co., Ltd., Shijiazhuang 050000, China; 4Animal Husbandry Development and Service Center in Jimo District, Qingdao 266000, China

**Keywords:** HP-PRRSV, recombinant virus, vaccine

## Abstract

A new HP-PRRSV strain (SD2020) was isolated from pigs with suspected highly pathogenic porcine reproductive and respiratory syndrome disease in a pig farm in Shandong Province, China, and its genome was sequenced. This pig farm has been using the VR-2332 vaccine strain to immunize pigs for a long time. The phylogenic and single nucleotide polymorphisms (SNPs) analysis of the viruses isolated from dead pigs showed that SD2020 was a natural recombinant virus of the VR-2332 vaccine strain and the JXA1 similar strain, and that two splicing fragments highly homologous to JXA1 in the virus genome were probably derived from the JXA1 wild strain and JXA1-R vaccine strain, respectively. Therefore, the possible recombination events of SD2020 and its mutation site might be related to high pathogenicity.

## 1. Introduction

Porcine reproductive and respiratory syndrome (PRRS), caused by the PRRS virus (PRRSV), brings about huge economic losses to global pig industry. The major manifestation of PRRS is a respiratory syndrome in sows with reproductive disorders. PPRRSV is a worldwide epidemic virus belonging to *Arteriviridae* family with its genotype 1 first identified in Europe (1991) and genetype 2 in America (1992) [1]. Highly pathogenic PRRS, characterized by high morbidity and mortality in pigs at all ages, first emerged in China in 2006, and it has been spreading [2]. Similar to JXA1, HuN4 strain, a unique discontinuous 30 amino acid (482 aa, 534–562 aa) deletion of nonstructural protein 2 (NSP2) was found in this strain [3]. Of note, emerging evidence shows that the whole virus gene pool of PRRSV becomes increasingly complex due to continuous outbreak of new virus strains, wild-type virus strain point mutations, and genome recombination [4,5,6].

Vaccination is the most effective and commonly used strategy to protect farmed animals from a variety of animal infectious diseases. Since the PRRSV-2 modified-live virus (MLV) vaccine was licensed in the United States in 1994 and PRRSV-1 MLV vaccine in Europe in 2000 [7], PRRS vaccine has been widely used in the pig industry all over the world, which consequently results in huge economic benefits. At present, there are two main types of vaccines, MLV vaccine and killed virus (KV) vaccine [8,9]. Although KV vaccine is safer, the MLV vaccine can provide better protection [10]. Therefore, MLV vaccines have been widely used in China, Among the MLV vaccines, VR-2332 and JXA1-R which belong to different lineages, are the most common vaccine strains [11]. However, the effectiveness of heterologous cross-protection and biosafety of the MLV vaccine are considered as two main obstacles for the application and development of the MLV vaccine. Among them, the biosafety risk caused by the shedding of vaccine virus strains including accelerated mutation and recombination has not been adequately evaluated. The strain HP-PRRSV has gradually become the mainstream strain in Chinese pig farms since 2006 [12,13]. To prevent and control HP-PRRSV, a number of Chinese pig farms use attenuated vaccines made from HP-PRRSV, such as JXA1-R. JXA1-R-attenuated vaccine showed preventive effect against HP-PRRSV. Unfortunately, it has been found that many live attenuated vaccines are able to recombine with field strains, thereby leading to an increasingly complex viral gene pool and difficulties in the control of the virus [5,14]. Current vaccines against HP-PRRSV, either MLV against PRSSV2 or JXA1-R, emerged to recombine with wild strains in the field, making it more difficult to control PRRSV [6].

In this study, we obtained one new full-length PRRSV genome from a pig farm in Shandong province, China. Retrospective data showed that the porcine in this farm was first immunized with the PRRS vaccine, porcine reproductive and respiratory syndrome vaccine live (CH-1R strain) before September 2019, followed by immunization with a different PRRS Vaccine, porcine reproductive and respiratory syndrome vaccine live (strain R98). However, it occurred that there were increasing numbers of nursery pigs with high fever (40.5–42 °C), dyspnea, and depression, with a mortality rate of 60%. We found that the newly identified virus (referred as SD2020) belonged to the PRRSV-2 type, which was naturally recombined from different lineage strains. Of note, the genome of this infectious strain was recombined from two vaccine strains with low pathogenicity. One of the vaccine strains was VR-2332, which had been used in the pig farm for a long time. We also found that the two recombinant regions were highly homologous to the JAX1-R vaccine strain, but some SNPs exhibited the characteristics of wild strains with high pathogenicity. The main regions of the genetic sequence of SD2020 strain are still VR-2332 vaccine strain, and one of the two recombinant regions is also suspected to be a vaccine strain, JXA1P80 vaccine strain. It is well known that vaccine strains in the vaccinated animals do not cause serious illness and death, that are the unique virulence consequences of these highly pathogenic strains. However, the SD2020 strain showed the characteristics of high pathogenicity, and one region of the SD2020 strain was identical to wild type SNP of JXA1 with high pathogenicity. Therefore, we speculated that these SNPs are major contributors to the high pathogenicity of the SD2020 strain. While the amino acid sequences of these genes are currently found to be different compared to those of HP-PRRSV, this strain does not have the typical mutational deletions characteristic of HP-PRRSV found previously before this time. We believe that these SNP alterations provide direction for our search for the cause of elevated pathogenicity of PRRSV. In contrast, these gene sequences deviate from the amino acid sequence of conventional HP-PRRSV and are not characterized by the typical mutational deletions of previous HP-PRRSV. In addition, we presume that the gene fragment that recombines with VR2332 is the key gene fragment responsible for the increased pathogenicity of PRRSV. Analysis of the gene sequence of this newly identified strain provides new directions for our investigation of the key genes responsible for the increased pathogenicity of the virus.

## 2. Materials and Methods

### 2.1. Sample Collection, Virus Isolation, Identification and Purification

The lung samples were collected from diseased pigs and prepared for virus isolation and histopathological studies. In this study, PRRSV was isolated by culturing in MARC-145 as previously described [15]. MARC-145 cells were cultured in Dulbecco’s Modified Eagle Medium (DMEM, Gibco, New York, NY, USA) supplemented with 10% fetal bovine serum (FBS, Gibco, New York, NY, USA) in a humidified incubator with 5% CO_2_ at 37 °C. The lung samples were added when Marc-145 cells reached a monolayer, after 1 h of adsorption in an incubator with 5% CO_2_ at 37 °C, the unattached virus was washed off and fresh DMEM with 2% FBS was added. Cells was were split through three blind serial passages until the appearance of cytopathic effects (CPE).

To purify PRRSV, plaque isolation was performed in Marc-145 cells for purification of virus particles. The frozen viral solution was removed and diluted in a gradient with DMEM containing 2% FBS in multiples of 10. When the fusion of Marc-145 cells in the 12-well plate reached 80% confluence, the diluted virus solution was added and adsorbed for 2 h. The virus solution was removed and DMEM mixed 1:1 with low melting point agarose was added. The DMEM contained 0.2% crystalline violet and 2% FBS. The cells were incubated in the incubator for 1 h and then inverted onto 12-well plates. The empty spots were removed and observed after 48 h, and independent empty spots were aspirated with a pipette tip. The viral solution was frozen after two blind passages in Marc-145 cells.

JXA1 (isolated and maintained in our laboratory) was used as a positive control to identify if PRRSV was successfully isolated. Isolated virus, as well as RNA from JXA1, was obtained by MiniBEST Viral RNA/DNA Extraction Kit Ver.5.0 (Takara, Dalian, China), as described in the instructions. The extracted genomic RNA was validated by PrimeScript^TM^ RT-PCR Kit (Takara, Dalian, China), with the following three steps: (1) Denaturation, annealing reaction: The reaction solution was prepared as mixture of dNTP Mixture (10 mM each) 1 µL, Random 6 mers (20 µM) 1 µL, Template RNA 3 µL and RNase Free dH_2_O 5 µL. The reaction was incubated at 65 °C for 5 min and terminated at 4 °C. (2) Reverse transcription reaction: The above denaturing and annealing reaction solution 10 µL, 5×PrimeScript Buffer 4 µL, Inhibitor (40 U/µL) 0.5 µL, Prime Script RTase (for 2 Step) 0.5 µL, RNase Freed H_2_O 5 µL and Total Volume 20 µL, the reaction was incubated at 30 °C for 10 min, 42 °C for 30 min, 70 °C for 15 min and 4 °C terminated. (3) RT-PCR reaction: 0×PCR BufferⅡ 5 µL, dNTP Mixture (10 mM each) 2 µL, ORF7-1 (10 μM) 0.5 µL, ORF7-2 (10 μM) 0.5 µL, TaKaRa Ex Taq HS (5 U/µL) 0.5 µL, Reverse transcription reaction solution as described above 2.5 µL and H_2_O 39 µL; the reaction was incubated at 95 °C for 3 min, 30 cycles of 95 °C for 30 s and 55 °C for 30 s and 72 °C for 1 min, 72 °C for 10 min and 4 °C terminated. Primers for PCR were designed to pass through the PRRSV N protein nucleotide sequence (ORF7-1 GCCCCTGCCCACCACG, ORF7-2 TCGCCCTAATTGA ATAGGTGA).

### 2.2. Growth Kinetic Curve

Viruses diluted 10 times by DMEM containing 2% FBS were inoculated in Marc-145 cells (80% confluence) in 96-well plates. The plates were incubated in the incubator for 96 h. The number of wells in which CPE appeared was observed, and the virus titer was calculated by the Reed–Muench rule.

SD2020 Multiplicity of infection (MOI) = 0.1 infected cell when cells were grown to approximately 80% confluency in a cell vial. After incubation at 37 °C for 1 h with rocking every 15 min, the virus suspension was discarded. Cells were cultured with DMEM containing 2% FBS in the incubator. Cells were removed and frozen at −20 °C at 24, 48, 72 and 96 h. Virus suspensions were collected by freeze-thawing the cells three times for the different incubation times.

The harvested virus suspension was diluted using DMEM containing 2% FBS, and the diluted virus solution was inoculated in Marc-145 cells (approximately 80%) in 96-well plates. Cells were incubated in the incubator for 96 h followed by evaluation of the number of CPE. The TCID_50_ was calculated using the Reed–Muench rule and the virus growth curve was plotted.

### 2.3. Pathological and Immunohistochemical Experiments

Specimens obtained from lung lesions of lung lobe were placed in 10% neutral buffered formalin and embedded in standard paraffin blocks, sectioned at 3 microns and stained with hematoxylin and eosin (HE). Specimens obtained from normal tissues were included as negative control. Immunohistochemistry (IHC) was used to confirm PRRS virus antigen. Tissue sections (3 microns) were dried in the air, dewaxed in xylene and rehydrated in decreasing strengths of alcohols (100% to 0%). After deparaffinization and rehydration, endogenous peroxidase activity was blocked with 3% H_2_O_2_ in methanol for 25 min, and the sections were then washed thrice in in Tris-buffered saline (TBS, 0.05 M, pH 7.6). For antigen retrieval, the sections were placed in citrate buffer (0.01 M, pH 6.0), heated in a microwave oven for 40 min, and cooled at room temperature. After washing with TBS, the sections were incubated with 5% bovine serum albumin (BSA) in a humidified chamber at 37 °C for 20 min to block non-specific binding sites in tissues. Sections were then incubated with a PRRSV monoclonal antibody (mouse anti PRRSV-N protein) at a dilution of 1:500 or non-immune mouse serum (control) for overnight at 4 °C. The sections were then thoroughly washed three times in TBS (5 min each), and incubated with HRP-labeled goat anti-mouse IgG (at a dilution of 1:100) at 37 °C for 1 h. After washing three times in TBS, the sections were incubated with diaminobenzidine tetrahydrochloride (DAB) and H_2_O_2_ at room temperature for 10 min followed by rinse with distilled water to stop the color reaction. Sections were counterstained with hematoxylin for 30 s and differentiated with 1% hydrochloric acid alcohol for 10 s. After dehydration with gradient alcohol and transparency in xylene, the sections were mounted with gum. Sections obtained from the normal lungs of healthy pig were stained as negative control.

### 2.4. Complete Virus Genome Sequencing

Viral RNA was extracted using a Viral RNA Extraction Kit (Takara, Dalian, China) according to the manufacturer’s instructions, followed by amplification using Prime Script^TM^ RT-PCR Kit (Takara, Dalian, China). The complete virus genome was amplified using 10 pair primers (Appendix A) and sequenced by Dalian Takara Co., Ltd. The complete s genome of the isolate was assembled using DNAMAN software.

### 2.5. Phylogenetic and Recombination Analyses

BLASTn was used to search related genes of the SD2020 virus strain, and the reference virus sequences were obtained from Genbank. Whole genome sequences of SD2020 and reference viruses were aligned using CLUSTALW [16]. To detect the probable recombination events, Simplot software v3.5.1 (a sliding window of 200-bp and step size of 20 bp) and RDP v4.1016 were used to scan the genomic sequences and performing boot scanning analysis with a sliding window of 200-bp (20-bp step size) [17,18]. To understand the evolutionary characterization of every region of SD2020, the neighbor-joining method with 1000 bootstraps of the aligned sequences for every recombination region was performed in the PHYLIP 3.65 software package [19].

### 2.6. SNP Analysis

With JXA1 (EF112445) and JXA1 P80 (FJ548853) as reference sequences, we analyzed B and D region sequence SNPs of SD2020, whole genome sequence SNPs of CH-YY (MK450365) and 15SC2 (KX815427), and whole genome sequence SNPs of vaccine rJXA1-R (MT163314) in the market.

## 3. Results

### 3.1. Pathological and Immunohistochemical Observation

Gross pathological observation results showed serious edema and hemorrhage in the lung (Figure 1A). The histopathological analysis exhibited typical pathology performance of viral acute lung injury, including a transparent membrane on the alveolar surface, emphysema, thickening of the interlobular septum, and infiltration of inflammatory cells within the pulmonary mesenchyme (Figure 1B,C). PRRSV antigens were detected in the lung tissues of diseased piglets by immunohistochemistry staining with monoclonal antibodies against N protein of PRRSV, and PRRSV positive signals were observed in pulmonary macrophage and intravascular macrophages (Figure 1D).

We successfully isolated the PRRSV strain using a plaque forming assay and the isolated strain was confirmed by RT-PCR experiments (Figure 2A,B).The virus growth curve showed that the replication efficiency of the virus gradually increased with increasing incubation time after virus inoculation. The replication efficiency of PRRSV peaked at 72 hpi of incubation and then began to decline. The 72 hpi of incubation was the time of the highest replication efficiency of SD2020. The replication efficiency of SD2020 was lower than that of JXA1. The multiplication profile of SD2020 was detected by growth kinetic experiments. It was able to observe that the maximum viral titer was measured at 72 hpi and decreased when the time exceeded 72 hpi. The TCID_50_ of SD2020 was log10^6.66^/mL and the TCID_50_ of JXA1 was log10^7.1^ TCID_50_/mL (Figure 2C).

### 3.2. Virus Isolation and Complete Genome Sequencing

Three virus strains from different lung tissue samples of fresh dead pigs were isolated by culturing and purifying them in MARC-145, Sequencing of whole virus gene fragments confirmed that these three strains had the same nucleic acid sequence. Viral RNA RT-PCR test results show that all the tested samples (3 virus strains and 32 typical lesion tissue samples) share the same genomic sequence. Therefore, we concluded that this PRRS outbreak was caused by a single virus strain infection. The virus strain was designated as SD2020, and its sequence full-length (GenBank accession no. MW408254) was 15386 bp.

### 3.3. Comparison with Other PRRSV Strains

Alignment of SD2020 and VR-2332 sequences revealed that the overall homology was 97.53%. Meanwhile, SD2020 shared 66.54% sequence identity with the Lelystad virus, indicating that SD2020 belongs to genotype 2 PRRSV. Furthermore, we compared the obtained genomes with the sequences of multiple virus genomes of the PRRSV-2 lineage that had been detected, including NADC30, MN184C, JA142, CH-1a, P129, HB-1, HB-2, JXA1, etc. The main sequence of SD2020 virus genome is highly homologous to the vaccine strain VR-2332 used in pig farms. Simultaneously, the partial sequence of the SD2020 virus genome had significantly higher homology with the JXA1 virus strain, than VR-2332. Based on these results, we speculated that SD2020 was a recombinant virus from the VR-2332 vaccine strain and JXA1-like strain.

Compared with the homologous region of the SD2020 strain and the VR2332 strain, there was a deletion of three amino acid residues in NSP2 protein (GVP, from position 593–595). The same pattern of NSP2 deletion has been found in both Chinese and American strains [20]. In addition, we did not identify insertions in NSP2 of SD2020.

### 3.4. Recombination and Phylogenetic in SD2020

The recombination analysis (similarity plot in SIMPLOT v3.5.1) supported that the SD2020 has a recombinant genome (Figure 3A). The RDP v4.1016 analysis identified four recombination sites (9698, 11,280, 12,828, and 14,351) (Figure 3C). The results revealed that the nucleotide sequences of site from 9698 to 11,280, and from 12,828 to 14,351 in the SD2020 genome were more closely related to JXA1, while the rest of the viral genome sequences were highly homologous to VR-2332.

To further analyze the evolutionary characteristics of SD2020 gene sequence, phylogenetic tree (neighbour-joining, 1,000 bootstrap replicates) was constructed for each region by MEGA X to locate SD2020′s evolutionary branch. According to recombination sites, we divided the whole SD2020 virus gene sequence into five regions, namely region A/B/C/D/E. Obviously, the phylogenetic analyses with separate regions show that regions A, C, and E were derived from the VR-2332 virus strain, whereas regions B and D were highly similar to the JXA1-like virus strains (Figure 3D). Given that the phylogenetic analysis was consistent with the recombination analysis, we conclude that the SD2020 strain was naturally recombined from the VR-2332 vaccine strain and the JXA1-like strain.

Interestingly, RDP analysis indicated that regions B and D were, respectively, from JXA1P80 (vaccine strain) and JXA1 (wild-type strain). Therefore, we believe that more in-depth analysis is necessary to trace the specific sources of different JXA1-like fragments in the SD2020 viral genome.

### 3.5. SNP Analysis Highlights Strain-Specific Differences

We searched for similar sequences with the B and D regions of SD2020 by BLASTn and found that CH-YY strains were highly homologous to SD2020 in these two regions, In addition, 15SC2 was highly homologous to the B region of SD2020. Further analysis and a literature review showed that both CH-YY and 15SC2 belonged to the JXA1 family without recombination to other sublineages [5].

With JXA1 (EF112445) and JXA1 P80 (FJ548853) as reference sequences, we analyzed B and D region sequence SNPs of SD2020, whole genome sequence SNPs of CH-YY (MK450365),15SC2(KX815427), and whole genome sequence SNPs of vaccine rJXA1-R(MT163314) in the market [21]. The results are presented in Appendix A.

Of the 97 SNPs that can distinguish JXA1 from JXA1P80, only 10 SNPs (scattered in the whole virus genome) of 15SC2 were the same as those of JXA1, the rest of the 87 SNPs belonged to the JXA1P80 type. Therefore, we speculated that 15SC2 was derived from the JXA1-P80 vaccine strain merely through point mutation, which was consistent with the previous study [5]. In addition, SNPs of CH-YY were more complex than those of 15SC [2]. Specifically, there were 52 JXA1-type SNPs and 39 JXA1P80-type SNPs. Moreover, SNPs from the same strain were clustered, which strongly indicated that CH-YY was highly likely to be a natural recombinant virus strain from JXA1 wild-type strain and JXA1P80 vaccine strain. In CH-YY, SNP 2533–4392 region (number of JXA1-type SNPs/that of JXA1-P80 SNPs was 15/2) and SNP11980–15293 region (JXA1 type/JXA1 P80 was 16/7) showed significant JXA1-type characteristics, while SNP390-2247 region (JXA1 type/JXA1 P80 is 3/14) and SNP4870-11510 region (JXA1 type/JXA1 P80 is 5/35) exhibited significant JXA1P80-type characteristics. SD2020′s B and D regions corresponding to CH-YY’s homologous fragments were located in different strain-type SNP regions. Specifically, region B was located in the JXA1P80-type SNP region and region D was located in JXA1-type SNP region.

In addition, a single-base insertion/deletion combination (involving three amino acid changes) and a 12-base insertion (involving four amino acid changes) could distinguish JXA1 from JXA1P80. Unlike JXA1, a G was inserted at bit 13,664 and a C was missing at bit 13,673 in JXA1P80. Similar mutations were observed in SD2020 and CH-YY, except that T was inserted at site 13664. In JXA1 P80, there was a 12-nucleotide insertion between C13693 and A13704. This 12-base insertion can be used as a genetic marker to distinguish vaccine strains from the wild-type strains. Of note, this 12-nucleotide insertion was not observed in SD2020 and CH-YY.

## 4. Discussion

From August to October in 2019, PRRS broke out in a pig farm in Shandong Province, China. Retrospective data have showed that the surveyed farm has been using VR-2332 vaccine to prevent PRRS for five years, without an outbreak of related disease. Unexpectedly, after the latest vaccination (Strain R98, live attenuated vaccine), all of the nursery pigs were diagnosed with high fever (41–42 °C), dyspnea, depression, and even death. The mortality rate was as high as 100%. The major symptoms of sick pigs included difficult breathing, high fever, abdominal respiration, and sitting like a dog, which consequently led to death from respiratory failure. The above-mentioned symptoms were evaluated to have the maximum gross-lung-lesion scores according to the previously reported standard scoring system. The analysis of PRRS pathogenesis in pig farm and pathological changes of dead pigs indicated that the PRRSV strain that caused the current PRRS outbreak was highly pathogenic.

After obtaining the full-length sequence of the virus genome, we found that the virus characterized by highly pathogenic PRRS has some very interesting characteristics. The sequence of the SD2020 virus genome is almost identical to the sequence of vaccine strains (VR-2332) used in the farm, and two regions (region B from 9698 to 11,280 site, region D from 12,828 to 14,351 site) of SD2020 are highly similar to JXA1-like virus strains. Therefore, the SD2020 virus strain obviously experienced the natural recombination of the vaccine strain and the wild strain, which makes it intriguing to identify the wild virus strain which was involved in the recombination with VR-2332 vaccine strain.

JXA1 virus strain was firstly isolated in Jiangxi Province, China in 2006 [2]. It has been prevalent since its discovery. JXA1 is a typical virulent strain of highly pathogenic PRRSV. In view of the unique immunogenicity of JXA1 and its great harm, researchers passaged JXA1 strain continuously in the laboratory to obtain the attenuated strain JXA1 P80, which assisted in developing a live attenuated JXA1-R vaccine [22]. Recent studies of JXA1-like recombinations and JXA1-R recombinations revealed a continuous epidemic of wild-type JXA1 virus and spread of JXA1-R vaccine virus to the surroundings, accompanied by wide cross-infection with various PRRSV strains in the host [5,6,23].

By searching for the virus strain with the highest homology to the B and D region sequence, we identified CH-YY strain (highly homologous to these two regions) and 15SC2 strain (highly homologous to B region). Both CH-YY and 15SC2, as naturally infected strains, have been isolated in Chinese mainland in recent years [5,24], In addition, CH-YY and 15SC2 belong to the JXA1 family without recombination to other sub-lineages.

SNP analysis technology has been widely used in genetic typing and base site mutation analysis related to the biological characteristics of many kinds of viruses, including influenza virus [25,26,27,28]. For example, Lan Ke et al. used the SNP-based mutation network structure to study the type of influenza A virus outbreak in 2009, and identified HA two SNP sites 658A and 1408T as the basis for early epidemic virus typing. At the same time, the possible changes of virus host selection caused by 658A mutation were speculated^C^. Similar SNP analysis techniques had also been used by Wu Bin et al. to trace the evolutionary path of human H5N1 influenza virus and speculate on the SNP sites that change the virulence of the virus [28]. SNP analysis technology has also been applied to the research field of PRRSV, but it is not widely used. Zen H Lu uses ultra-deep sequencing to obtain SNP data, which provides a technical basis for further exploring the micro-evolution events of PRRSV [29]. Tomasz Stadejek et al. have noticed that single nucleotide mutations will lead to the early appearance of the stop codon in the ORF7 of PRRSV-1, and the size of the nucleocapsid protein of the virus becomes smaller [1]. Therefore, we adopted SNP technology to trace the evolutionary path of PRRSV virus and mutation of key amino acid residues (missense mutation of nucleotide bases) related to virulence of the SD2020 virus.

Further SNP analysis revealed the more specific sources of the two viruses. Specifically, we analyzed all SNP between the JXA1 wild-type strain and JXA1P80 vaccine strain, and determined all SNP types of CH-YY and 15SC2 strains which were most genetically similar to SD2020 strain, and further conducted a parallel control analysis of the rJXA1-R strain genome (Appendix A). The results showed that 15SC2 evolved from the vaccine strain through point mutation, while CH-YY was probably derived from the natural recombination of JXA1 like wild type strain and vaccine strain. The partial sequence characteristics of rJXA1-R vaccine strain genome were the same as those of JXA1 wild-type strain, but different from those of JXA1P80 vaccine strain, which was identical with 11005C, 13963A, and 12 base insertions in region B and D of SD2020. Based on these results, it could be speculated that these gene polymorphisms may not be related to the pathogenicity of SD2020.

Related research was found that the pathogenicity of wild strain and vaccine strain was significantly enhanced after recombination [30,31]. Therefore, this study further clarified the B and D region source of SD2020 and explored possible reasons for SD2020’s high mortality through SNP analysis of CH-YY, 15SC2, and SD2020.

Combining SNP analysis with sequence recombination analysis results, we found that the main genetic marker features of region B and D were highly similar to those of CH-YY, indicating that they had a very close common ancestor. Based on this, the common ancestor virus strain was speculated to have the characteristics of different JXA1 virus families in different regions and to be a recombinant strain of JXA1 and JXA1-R virus strains. Furthermore, this common ancestor recombined with VR-2332 under natural conditions to generate the SD2020 strain with high morbidity and mortality. Regions B and D of SD2020 strains were derived from the JXA1-like highly pathogenic wild-type strain and safe vaccine strain, respectively. Region A/C/E, highly homologous to VR-2332, are typical vaccine sequences with low pathogenicity and high safety. Based on the above-mentioned analysis results, we speculated that the D region might be directly related to high pathogenicity of SD2020. The mutation sites in D region, which were same as those of wild-type but different from those of the vaccine strain, were likely to be the highly pathogenic characteristic sites of SD2020. Only 10725G, 13344A and 13418T belonged to this type of mutation site. These three mutation sites are suggested to be further investigated in order to better understand the biological mechanism of PRRSV pathogenicity.

In this research, we determined that SD2020 is a new natural recombinant virus strain through a variety of sequence analysis methods, and reasonably speculated the evolutionary origin and pathway of SD2020. At the same time, we speculate that three SNP sites may be related to the virulence of the virus, according to the biological characteristics and SNP characteristics of the virus. Our work uncovered some key gene or protein characteristics of highly pathogenic PRRSV. However, further experimental validation of this speculation is needed. In addition, the SNP site related to viral virulence requires further exploration.

## Figures and Tables

**Figure 1 viruses-15-00165-f001:**
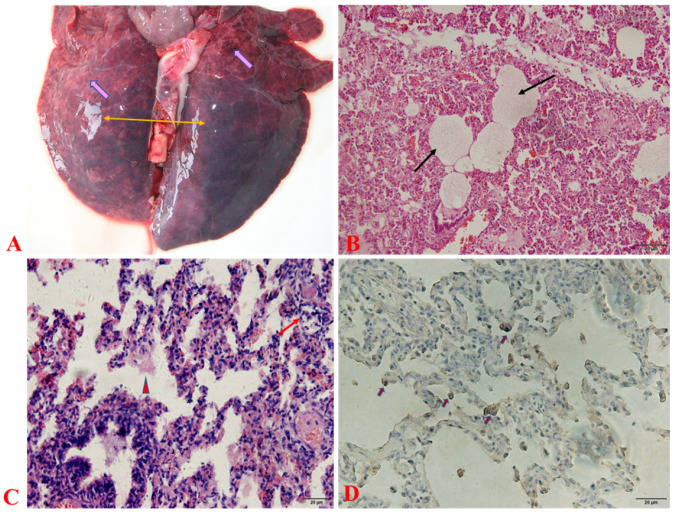
Pathological changes in infected lung tissue. (**A**) Gross lesions (**B**,**C**) Microscopic lesions (**D**) Antigens were detected by immunohistochemistry using monoclonal antibodies against N protein of PRRSV. “
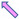
” indicates hemorrhage, “
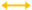
” indicates edema, “
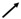
” indicates transparent membrane, “
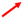
” indicates lymphocytic infiltration, “
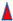
” indicates inflammatory exudation, “
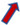
” indicates positive signal.

**Figure 2 viruses-15-00165-f002:**
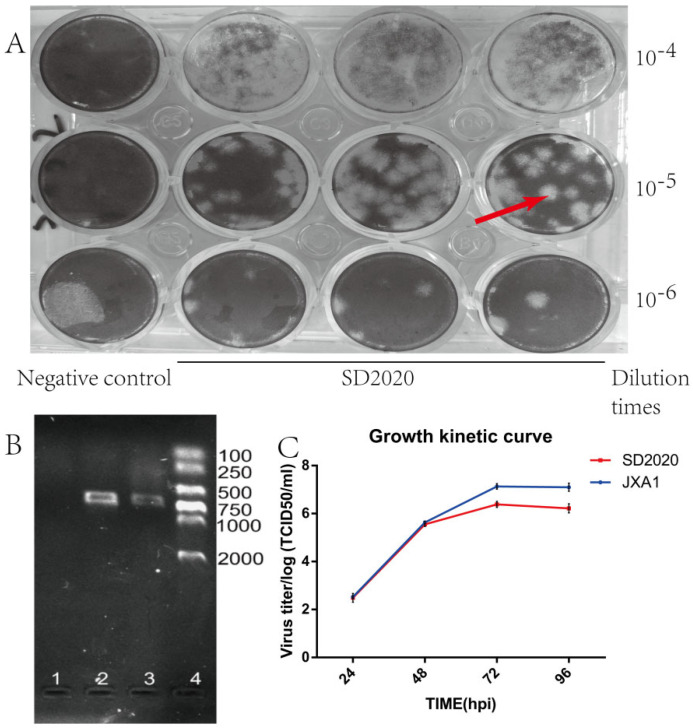
Virus isolation and identification. (**A**) SD2020 empty spot assay for isolation and purification, red arrows point to virus empty spots. (**B**) Isolation of SD2020 in Marc-145 cells and the result of RT−PCR assay. Lane 1 indicates the negative control, lane 2 indicates the positive control of JXA1, lane 3 indicates the result of SD2020 RT−PCR, and lane 4 indicates the DNA marker (DL 2000). (**C**) Growth kinetic curve of SD2020.

**Figure 3 viruses-15-00165-f003:**
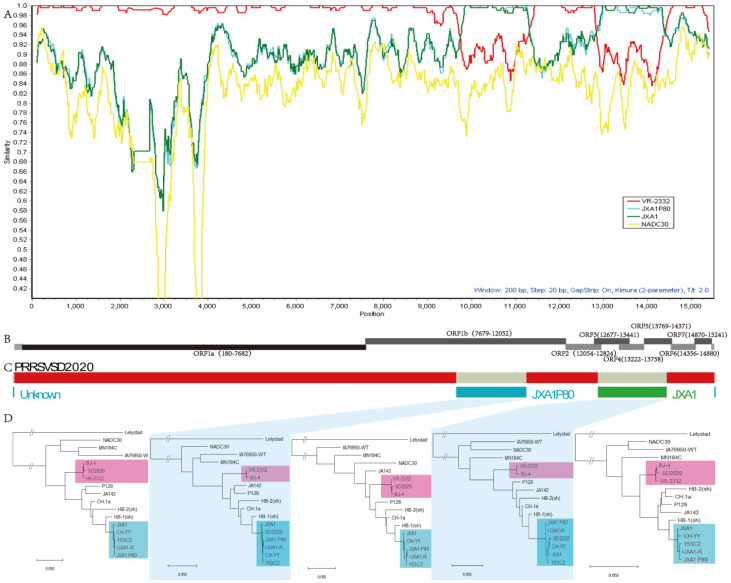
Recombination analyses on SD2020 genome. (**A**) Recombination analysis was conducted and a similarity map was generated by Simplot 3.5.1 software with window size of 200 bp and step size of 20 bp. SD2020 as query sequence, was used in similarity plot analysis. (**B**) Schematic diagram of SD2020 PRRSV genomes. (**C**) Recombination regions were shown with segments of different colors and the four recombination breakpoints were detected by RDP v4.1016. Red segments represent the sequence of VR-2332 strain. Blue segment represents the sequence of JXA1-P80 strain. Green segment represents the sequence of JXA1 strain. (**D**) Individual phylogenies reconstructed from non-recombinant fragments identified by RDP v4.1016, with the JXA1-like strain labelled with a blue background and the VR2332 strain labelled with a red background. The phylogenetic tree was constructed by using the neighbor-joining method with 1000 bootstrap replicates in the PHYLIP 3.65 software package.

## Data Availability

Not applicable.

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
