# Peer review of "Recombination and Mutation in a New HP-PRRSV Strain (SD2020) from China"

_viruses, 2023, doi:10.3390/v15010165_

Round 1

Reviewer 1 Report (New Reviewer)

The manuscript drafted by Li et al., investigate the possibility of recombination of two vaccine strains of PRRSV VR-2332 & JXA1 into a new HP-PRRSV strain (SD2020) by using the phylogenic and SNP analysis. The manuscript is generally well-referenced and logically organized, the results are clearly interpreted and discussed. To improve the manuscript, the reviewer hopes that the following comments and suggestions will be addressed.

Major concerns:

The authors analyzed the recombination and pathogenicity of SD2020 from the perspective of gene sequence analysis and identified the suspected SNP sites. However, there is no further animal studies on these SNP sites to support this speculation. This experimental study would be a strong justification for the conclusion.

Minor concerns:

1. L27-28: brings about huge economic losses to global pig industry. should be deleted, for it just repeats the previous sentence in Line 26.

2. L34: “Shows” should be changed to “show”.

3. L37: Delete are.

4. L44: NLV is a write error. “Which” should be changed to “which”.

5. L120 and 124: Some space symbols are missing.

6. Figure 1: annotations for various arrow symbols should be listed in the legend, not on the right of these panels.

7. L212: hour should be abbreviated as h or hpi, but not a mixture of the two.

8. All figures should be labeled with consecutive uppercase or lowercase letters, but not a mixture of the two.

9. Figure 2A: all annotations should not be a mixture of Chinese and English wording. The same problem exists in Figure 3A.

10. Figure 2B: flipping pictures is not allowed.

11. Figure 2C: it should be Virus titer/log (TCID50/ml) in the y-coordinate.

Author Response

Reviewer 2 Report (Previous Reviewer 1)

Dear respected editor, 

The revised version of this manuscript is acceptable for me. All needed corrections are done. I think it could be acceptable for publication in the current corrected form.

Author Response

We are very honoured that you have endorsed this manuscript. Thank you for all your hard work.

Reviewer 3 Report (Previous Reviewer 2)

The authors have made all the necessary changes to the comments mentioned in the previous version. 

Author Response

We are very honoured that you have endorsed this manuscript. Thank you for all your hard work.

This manuscript is a resubmission of an earlier submission. The following is a list of the peer review reports and author responses from that submission.

Round 1

Reviewer 1 Report

Dear editor of Viruses

I hope you are fine. Regarding the revision of the Manuscript No. viruses- 2001391, titled “Recombination of VR-2332 vaccine strain and JXA1-derived strains of PRRSV”. 

Some comments need to be replied.

Comments:

1-    Please indicate well the nucleotide deletions and insertions in the non-structural protein 2 (nsp2) of the detected recombined strain in your results. 

2-    Describe the lesions in Figure 1.

3-    The quality in Figure 2 needs to be improved especially the name and number in the phylogenetic trees (Fig. 2d).

4-    Add conclusion at the end of discussion.

Reviewer 2 Report

My comments are attached 
